# Interpreting Neural Network Judgments via Minimal, Stable, and Symbolic Corrections

**Xin Zhang**
CSAIL, MIT
xzhang@csail.mit.edu

**Armando Solar-Lezama**
CSAIL, MIT
asolar@csail.mit.edu

**Rishabh Singh**
Google Brain
rising@google.com

## Abstract

We present a new algorithm to generate minimal, stable, and symbolic corrections to an input that will cause a neural network with ReLU activations to change its output. We argue that such a correction is a useful way to provide feedback to a user when the network's output is different from a desired output. Our algorithm generates such a correction by solving a series of linear constraint satisfaction problems. The technique is evaluated on three neural network models: one predicting whether an applicant will pay a mortgage, one predicting whether a first-order theorem can be proved efficiently by a solver using certain heuristics, and the final one judging whether a drawing is an accurate rendition of a canonical drawing of a cat.

## 1   Introduction

When machine learning is used to make decisions about people in the real world, it is extremely important to be able to explain the rationale behind those decisions. Unfortunately, for systems based on deep learning, it is often not even clear what an explanation means; showing someone the sequence of operations that computed a decision provides little actionable insight. There have been some recent advances towards making deep neural networks more interpretable (e.g. [21]) using two main approaches: i) generating input prototypes that are representative of abstract concepts corresponding to different classes [23] and ii) explaining network decisions by computing relevance scores to different input features [1]. However, these explanations do not provide direct actionable insights regarding how to cause the prediction to move from an undesirable class to a desirable class.

In this paper, we argue that for the specific class of *judgment problems*, minimal, stable, and symbolic corrections are an ideal way of explaining a neural network decision. We use the term judgment to refer to a particular kind of binary decision problem where a user presents some information to an algorithm that is supposed to pass judgment on its input. The distinguishing feature of judgments relative to other kinds of decision problems is that they are asymmetric; if I apply for a loan and I get the loan, I am satisfied, and do not particularly care for an explanation; even the bank may not care as long as on aggregate the algorithm makes the bank money. On the other hand, I very much care if the algorithm denies my mortgage application. The same is true for a variety of problems, from college admissions, to parole, to hiring decisions. In each of these cases, the user expects a positive judgment, and would like an actionable explanation to accompany a negative judgment.

We argue that a *correction* is a useful form of feedback; what could I have done differently to elicit a positive judgment? For example, if I applied for a mortgage, knowing that I would have gotten a positive judgment if my debt to income ratio (DTI) was 10% lower is extremely useful; it is actionable information that I can use to adjust my finances. We argue, however, that the most useful corrections are those that are minimal, stable and symbolic.

First, in order for a correction to be actionable, the corrected input should be as similar as possible from the original offending input. For example, knowing that a lower DTI would have given me the

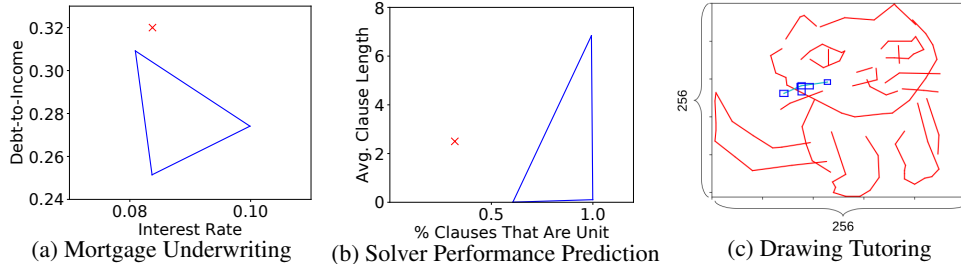

(a) Mortgage Underwriting          (b) Solver Performance Prediction          (c) Drawing Tutoring

Figure 1: Symbolic explanations generated by our approach for neural networks in different domains.

loan is useful, but knowing that a 65 year old billionaire from Nebraska would have gotten the loan is not useful. Minimality must be defined in terms of an error model which specifies which inputs are subject to change and how. For a bank loan, for example, debt, income and loan amount are subject to change within certain bounds, but I will not move to another state just to satisfy the bank.

Second, the suggested correction should be stable, meaning that there should be a neighborhood of points surrounding the suggested correction for which the outcome is also positive. For example, if the algorithm tells me that a 10% lower DTI would have gotten me the mortgage, and then six months later I come back with a DTI that is 11% lower, I expect to get the mortgage, and will be extremely disappointed if the bank says, "oh, sorry, we said 10% lower, not 11% lower". So even though for the neural network it may be perfectly reasonable to give positive judgments to isolated points surrounded by points that get negative judgments, corrections that lead to such isolated points will not be useful.

Finally, even if the correction is minimal and robust, it is even better if rather than a single point, the algorithm can produce a *symbolic* correction that provides some insight about the relationship between different variables. For example, knowing that for someone like me the bank expects a DTI of between 20% and 30% is more useful than just knowing a single value. And knowing something about how that range would change as a function of my credit score would be even more useful still.

In this paper, we present the first algorithm capable of computing minimal stable symbolic corrections. Given a neural network with ReLU activations and an input with a negative judgment, our algorithm produces a symbolic description of a space of corrections such that any correction in that space is guaranteed to change the judgment. In the limit, the algorithm will find the closest region with a volume above a given threshold. Internally, our algorithm reduces the problem into a series of linear constraint satisfaction problems, which are solved using the Gurobi linear programming (LP) solver [9]. We show that in practice, the algorithm is able to find good symbolic corrections in 12 minutes on average for small but realistic networks. While the running time is dominated by solver invocations, only under 2% of it is spent is spent on actual solving and the majority of the time is spent on creating these LP instances. We evaluate our approach on three neural networks: one predicting whether an applicant will pay a mortgage, one predicting whether a first-order theorem can be proved efficiently by a solver using certain heuristics, and the other judging whether a drawing is an accurate rendition of a canonical drawing of a cat.

**Explanation showcases.** Figure 1 shows example explanations generated by our approach on the aforementioned networks. Figure 1(a) suggests a mortgage applicant to change DTI and interest rate in order to get their application accepted. While the red cross represents the original application, the blue triangle represents the symbolic correction (i.e. the region of points that all lead to a positive outcome). Since the user may only be able to change DTI and interest rates often vary between applications, it is essential to provide a symbolic correction rather than a concrete correction to make the feedback actionable. Figure 1(b) suggests a user to reformulate a first-order theorem when the network predicts it as challenging to solve. Intuitively, either reducing the problem size (by decreasing average clauses lengths) or providing a partial solution (by adding unit clauses) would reduce the problem complexity. Finally, Figure 1(c) shows how to add lines to a drawing so that it gets recognized by the network as a canonical cat drawing. The red lines represent the original input, while the blue boxes represent the symbolic correction and the cyan lines represent one concrete correction in it. Briefly, any concrete correction whose vertices fall into the blue boxes would make the drawing pass the network's judgment. Comparing to the previous two corrections which only involves 2 features, this correction involves 8 features (coordinates of each vertex) and can go upto 20 features. This highlights our approach's ability to generate relatively complex corrections.

## 2  Background and Problem Definition

We first introduce some notations we will use in explaining our approach. Suppose $F$ is a neural network with ReLU activation. In the model we consider, the input to $F$ is a (column) vector $\boldsymbol{v}_0$ [1] of size $s_0$. The network computes the output of each ReLU (hidden or output) layer as

$$\boldsymbol{v}_{i+1} = f_i(\boldsymbol{v}_i) = \mathsf{ReLU}(\boldsymbol{W}_i \boldsymbol{v}_i + \boldsymbol{b}_i)$$

Where $\boldsymbol{W}_i$ is an $s_{i+1} \times s_i$ matrix, $\boldsymbol{b}_i$ is a vector of size $s_{i+1}$, and ReLU applies the rectifier function elementwise to the output of the linear operations.

We focus on classification problems, where the classification of input $\boldsymbol{v}$ is obtained by

$$l_F(\boldsymbol{v}) \in \mathsf{argmax}_i F(\boldsymbol{v})[i].$$

We are specifically focused on binary classification problems (that is, $l_F(\boldsymbol{v}) \in \{0, 1\}$). The *judgment problem* is a special binary classification problem where one label is preferable than the other. We assume 1 is preferable throughout the paper.

The *judgment interpretation problem* concerns providing feedback in the form of corrections when $l_F(\boldsymbol{v}) = 0$. A correction $\boldsymbol{\delta}$ is a real vector of input vector length such that $l_F(\boldsymbol{v} + \boldsymbol{\delta}) = 1$. As mentioned previously, a desirable feedback should be a minimal, stable, and symbolic correction. We first introduce what it means for a concrete correction $\boldsymbol{\delta}$ to be minimal and stable. Minimality is defined in terms of a norm $\|\boldsymbol{\delta}\|$ on $\boldsymbol{\delta}$ that measures the distance between the corrected input and the original input. For simplicity, we use $L_1$ norm to measure the sizes of all vectors throughout Section 2 and Section 3. We say $\boldsymbol{\delta}$ is $e$-stable if for any $\boldsymbol{\delta}'$ such that if $\|\boldsymbol{\delta} - \boldsymbol{\delta}'\| \leq e$, we have $l_F(\boldsymbol{v} + \boldsymbol{\delta}') = 1$.

A symbolic correction $\boldsymbol{\Delta}$ is a connected set of concrete corrections. More concretely, we will use a set of linear constraints to represent a symbolic correction. We say a symbolic correction is $e$-stable if there exists a correction $\boldsymbol{\delta} \in \boldsymbol{\Delta}$ such that for any $\boldsymbol{\delta}'$ where $\|\boldsymbol{\delta}' - \boldsymbol{\delta}\| \leq e$, we have $\boldsymbol{\delta}' \in \boldsymbol{\Delta}$. We call such a correction a stable region center inside $\boldsymbol{\Delta}$. To define minimality, we define the distance of $\boldsymbol{\Delta}$ from the original input using the distance of a stable region center that has the smallest distance among all stable region centers. More formally:

$$\mathsf{dis}_e(\boldsymbol{\Delta}) := \mathsf{min}_{\boldsymbol{\delta} \in S} \|\boldsymbol{\delta}\|,$$

where $S := \{\boldsymbol{\delta} \in \boldsymbol{\Delta} \mid \forall \boldsymbol{\delta}'.\|\boldsymbol{\delta}' - \boldsymbol{\delta}\| \leq e \implies \boldsymbol{\delta}' \in \boldsymbol{\Delta}\}$. When $\boldsymbol{\Delta}$ is not $e$-stable, $S$ will be empty, so we define $\mathsf{dis}_e(\boldsymbol{\Delta}) := \infty$.

We can now define the judgment interpretation problem.

**Definition 1.** (*Judgment Interpretation*) Given a neural network $F$, an input vector $\boldsymbol{v}$ such that $l_F(\boldsymbol{v}) = 0$, and a real value $e$, a judgment interpretation is an $e$-stable symbolic correction $\boldsymbol{\Delta}$ with the minimum distance among all $e$-stable symbolic corrections.

## 3  Our Approach

Algorithm 1 outlines our approach to find a judgment interpretation for a given neural network $F$ and an input vector $\boldsymbol{v}$. Besides these two inputs, it is parameterized by a real $e$ and an integer $n$. The former specifies the radius parameter in our stability definition, while the latter specifies how many features are allowed to vary to produce the judgment interpretation. We parameterize the number of features to change as high-dimension interpretations can be hard for end users to understand. For instance, it is very easy for a user to understand if the explanation says their mortgage would be approved as long as they change the DTI and the credit score while keeping the other features as they were. On the other hand, it is much harder to understand an an interpretation that involves all features (in our experiment, there are 21 features for the mortgage underwriting domain). The output is a judgment interpretation that is expressed in a system of linear constraints, which are in the form of

$$\boldsymbol{A}\boldsymbol{x} + \boldsymbol{b} \geq 0,$$

where $\boldsymbol{x}$ is a vector of variables, $\boldsymbol{A}$ is a matrix, and $\boldsymbol{b}$ is a vector.

Algorithm 1 finds such an interpretation by iteratively invoking findProjectedInterpretation (Algorithm 2) to find an interpretation that varies a list of $n$ features $\boldsymbol{s}$. It returns the one with the least

**Algorithm 1** Finding a judgment interpretation.

**INPUT** A neural network $F$ and an input vector $\boldsymbol{v}$ such that $l_F(\boldsymbol{v}) = 0$.
**OUTPUT** A judgment interpretation $\boldsymbol{\Delta}$.
1: **PARAM** A real value $e$ and an integer number $n$.
2: $\boldsymbol{S_n} := \{ \boldsymbol{s} \mid \boldsymbol{s}$ is a subarray of $[1, ..., |\boldsymbol{v}|]$ with length $n \}$
3: $\boldsymbol{\Delta} := None, d := +\infty$
4: **for** $\boldsymbol{s} \in \boldsymbol{S_n}$ **do**
5: $\quad \boldsymbol{\Delta}_s :=$ findProjectedInterpretation$(F, \boldsymbol{v}, \boldsymbol{s}, e)$
6: $\quad$ **if** $\mathsf{dis}_e(\boldsymbol{\Delta}_s) < d$ **then**
7: $\quad\quad \boldsymbol{\Delta} := \boldsymbol{\Delta}_s, d := \mathsf{dis}_e(\boldsymbol{\Delta}_s)$
8: **return** $\boldsymbol{\Delta}$

**Algorithm 2** findProjectedInterpretation

**INPUT** A neural network $F$, an input vector $\boldsymbol{v}$, an integer vector $\boldsymbol{s}$, and a real number $e$.
**OUTPUT** A symbolic correction $\boldsymbol{\Delta}_s$ that only changes features indexed by $\boldsymbol{s}$.
1: **PARAM** An integer $m$, the maximum number of verified linear regions to consider.
2: regions $:= \emptyset$, workList $:= []$
3: $\boldsymbol{\delta}_0 :=$ findMinimumConcreteCorrection$(F, \boldsymbol{v}, \boldsymbol{s})$
4: $\boldsymbol{a}_0 :=$ getActivations$(F, \boldsymbol{\delta}_0 + \boldsymbol{v})$
5: $L_0 :=$ getRegionFromActivations$(F, \boldsymbol{a}_0, \boldsymbol{v}, \boldsymbol{s})$
6: regions $:=$ regions $\cup \{L_0\}$
7: workList $:=$ append(workList, $\boldsymbol{a}_0$)
8: **while** len(workList)$! = 0$ **do**
9: $\quad \boldsymbol{a} :=$ popHead(workList)
10: $\quad$ **for** $p \in [1, \mathsf{len}(\boldsymbol{a})]$ **do**
11: $\quad\quad$ **if** checkRegionBoundary$(F, \boldsymbol{a}, p, \boldsymbol{v}, \boldsymbol{s})$ **then**
12: $\quad\quad\quad \boldsymbol{a'} :=$ copy$(\boldsymbol{a})$
13: $\quad\quad\quad \boldsymbol{a'}[p] := \neg \boldsymbol{a'}[p]$
14: $\quad\quad\quad L' :=$ getRegionFromActivations$(F, \boldsymbol{a'}, \boldsymbol{v}, \boldsymbol{s})$
15: $\quad\quad\quad$ **if** $L' \notin$ regions **then**
16: $\quad\quad\quad\quad$ regions $:=$ regions $\cup \{L'\}$
17: $\quad\quad\quad\quad$ **if** len(regions) $= m$ **then**
18: $\quad\quad\quad\quad\quad$ workList $:= []$
19: $\quad\quad\quad\quad\quad$ **break**
20: $\quad\quad\quad\quad$ workList $:=$ append(workList, $\boldsymbol{a'}$)
21: **return** inferConvexCorrection(regions)

distance. Recall that the distance is defined as $\mathsf{dis}_e(\boldsymbol{\Delta}) = \min_{\boldsymbol{\delta} \in S} \|\boldsymbol{\delta}\|$, which can be evaluated by solving a sequence of linear programming problems when $L_1$ norm is used.

We next discuss findProjectedInterpretation which is the heart of our approach.

## 3.1 Finding a Judgment Interpretation along given features

In order to find a judgment interpretation, we need to find a set of linear constraints that are **minimal**, **stable**, and **verified** (that is, all corrections satisfying it will make the input classified as 1). None of these properties are trivial to satisfy given the complexity of any real-world neural network.

We first discuss how we address these challenges at a high level, then dive into the details of the algorithm. To address minimality, we first find a single concrete correction that is minimum by leveraging an existing adversarial example generation technique [7] and then generate a symbolic correction by expanding upon it. To generate a stable and verified correction, we exploit the fact that ReLU-based neural networks are piece-wise linear functions. Briefly, all the inputs that activate the same set of neurons can be characterized by a set of linear constraints. We can further characterize the subset of inputs that are classified as 1 by adding an additional linear constraint. Therefore, we can use a set of linear constraints to represent a set of verified concrete corrections under certain activations. We call this set of corrections a *verified linear region* (or *region* for short). We first identify the region that the initial concrete correction belongs to, then grow the set of regions by identifying regions that are connected to existing regions. Finally, we infer a set of linear constraints whose concrete corrections are a subset of ones enclosed by the set of discovered regions. Algorithm 2 details our approach, which we describe below.

**Generating the initial region.** We first find a minimum concrete correction $\boldsymbol{\delta}_0$ by leveraging a modified version of the fast signed gradient method [7] that minimizes the $L_1$ distance (on line 3). More concretely, starting with a vector of 0s, we calculate $\boldsymbol{\delta}_0$ by iteratively adding a modified gradient that takes the sign of the most significant dimension among the selected features until $l_F(\boldsymbol{v} + \boldsymbol{\delta}_0) = 1$. For example, if the original gradient is $[0.5, 1.0, 6.0, -6.0]$, the modified gradient would be $[0, 0, 1.0, 0]$ or $[0, 0, 0, -1.0]$. Then we obtain the ReLU activations $\boldsymbol{a_0}$ for $\boldsymbol{v} + \boldsymbol{\delta}_0$ (by invoking getActivations on line 4), which is a Boolean vector where each Boolean value represents

whether a given neuron is activated. Finally, we obtain the initial region that $\boldsymbol{\delta}_0$ falls into by invoking getRegionFromActivations (on line 5), which is defined below:

$$\text{getRegionFromActivations}(F, \boldsymbol{a}, \boldsymbol{v}, \boldsymbol{s}) := \text{activationConstraints}(F, \boldsymbol{a}, \boldsymbol{v}) \wedge \text{classConstraints}(F, \boldsymbol{a}, \boldsymbol{v})$$
$$\wedge \text{featureConstraints}(\boldsymbol{s}),$$

where $\text{activationConstraints}(F, \boldsymbol{a}, \boldsymbol{v}) := \quad \bigwedge_{j \in [1,k]} \bigwedge_{m \in [1,|f_j|]} \{G_r^{\boldsymbol{a}}(\boldsymbol{x} + \boldsymbol{v}) \geq 0 \text{ if } \boldsymbol{a}[r] = \text{true}\}$

$$\wedge \bigwedge_{j \in [1,k]} \bigwedge_{m \in [1,|f_j|]} \{G_r^{\boldsymbol{a}}(\boldsymbol{x} + \boldsymbol{v}) < 0 \text{ if } \boldsymbol{a}[r] = \text{false}\},$$
$$\text{where } G_r^{\boldsymbol{a}}(\boldsymbol{x} + \boldsymbol{v}) := \boldsymbol{w}_r \cdot f_{m-1}^{\boldsymbol{a}}(...f_1^{\boldsymbol{a}}(f_0^{\boldsymbol{a}}(\boldsymbol{x} + \boldsymbol{v}))) + b_r,$$
$$r := \sum_{i \in [1,j-1]} |f_i| + m$$

$$\text{classConstraints}(F, \boldsymbol{a}, \boldsymbol{v}) := F^{\boldsymbol{a}}(\boldsymbol{x} + \boldsymbol{v})[1] > F^{\boldsymbol{a}}(\boldsymbol{x} + \boldsymbol{v})[0],$$

$$\text{featureConstraints}(\boldsymbol{s}) := \bigwedge_{j \notin \boldsymbol{s}} \boldsymbol{x}[j] = 0.$$

In the definition above, we use the notation $f_i^{\boldsymbol{a}}$ to refer to layer $i$ with its activations "fixed" to $\boldsymbol{a}$. More formally, $f_i^{\boldsymbol{a}}(\boldsymbol{v}_i) = \boldsymbol{W}_i^{\boldsymbol{a}} \boldsymbol{v}_i + \boldsymbol{b}_i^{\boldsymbol{a}}$ where $\boldsymbol{W}_i^{\boldsymbol{a}}$ and $\boldsymbol{b}_i^{\boldsymbol{a}}$ have zeros in all the rows where the activation indicated that rectifier in the original layer had produced a zero. We use $k$ to represent the number of ReLU layers and $|f_j|$ to represent the number of neurons in the $j$th layer. Integer $r$ indexes the $m$th neuron in $j$th layer. Vector $\boldsymbol{w}_r$ and real number $b_r$ are the weights and the bias of neuron $r$ respectively. Intuitively, activationConstraints uses a set of linear constraints to encode the activation of each neuron.

**Expanding to connecting regions.** After generating the initial region, Algorithm 1 tries to grow the set of concrete corrections by identifying regions that are connected to existing regions (line 6-20). How do we know whether a region is connected to another efficiently? There are $2^n$ regions for a network with $n$ neurons and checking whether two sets of linear constraints intersect can be expensive on high dimensions. Intuitively, two regions are likely connected if their activations only differ by one ReLU. However, this is not entirely correct given a region is not only constrained by the activations by also the desired classification.

Our key insight is that, *since a ReLU-based neural network is a continuous function, two regions are connected if their activations differ by one neuron, and there are concrete corrections on the face of one of the corresponding convex hulls, and this face corresponds to the differing neuron.* Intuitively, on the piece-wise function represented by a neural network, the sets of concrete corrections in two adjacent linear pieces are connected if there are concrete corrections on the boundary between them. Following the intuition, we define checkRegionBoundary:

$$\text{checkRegionBoundary}(F, \boldsymbol{a}, p, \boldsymbol{v}, \boldsymbol{s}) := \text{isFeasible}(\text{boundaryConstraints}(F, \boldsymbol{a}, \boldsymbol{v}, p)$$
$$\wedge \text{classConstraints}(F, \boldsymbol{a}, \boldsymbol{v}) \wedge \text{featureConstraints}(\boldsymbol{s}))$$

where

$$\text{boundaryConstraints}(F, \boldsymbol{a}, p, \boldsymbol{v}) := \quad \bigwedge_{j \in [1,k]} \bigwedge_{m \in [1,|f_j|]} \{G_r^{\boldsymbol{a}}(\boldsymbol{x} + \boldsymbol{v}) = 0 \text{ if } r = p\}$$
$$\wedge \bigwedge_{j \in [1,k]} \bigwedge_{m \in [1,|f_j|]} \{G_r^{\boldsymbol{a}}(\boldsymbol{x} + \boldsymbol{v}) \geq 0 \text{ if } \boldsymbol{a}[r] = \text{true and } r! = p\}$$
$$\wedge \bigwedge_{j \in [1,k]} \bigwedge_{m \in [1,|f_j|]} \{G_r^{\boldsymbol{a}}(\boldsymbol{x} + \boldsymbol{v}) < 0 \text{ if } \boldsymbol{a}[r] = \text{false and } r! = p\}$$

where $G_r^{\boldsymbol{a}}(\boldsymbol{x} + \boldsymbol{v}) := \boldsymbol{w}_r \cdot f_{m-1}^{\boldsymbol{a}}(...f_1^{\boldsymbol{a}}(f_0^{\boldsymbol{a}}(\boldsymbol{x} + \boldsymbol{v}))) + b_r$ and $r := \sum_{i \in [1,j-1]} |f_i| + m$.

By leveraging checkRegionBoundary, Algorithm 2 uses a worklist algorithm to identify regions that are connected or transitively connected to the initial region until no more such regions can be found or the number of discovered regions reaches a predefined upper bound $m$ (line 8-20).

**Infer the final explanation.** Finally, Algorithm 2 infers a set of linear constraints whose corresponding concrete corrections are contained in the discovered regions. Moreover, to satisfy the stability constraint, we want this set to be as large as possible. Intuitively, we want to find a convex hull (represented by the returning constraints) that is contained in a polytope (represented by the regions), such that the volume of the convex hull is maximized. Further, we infer constraints that represent relatively simple shapes, such as simplexes or boxes, for two reasons. First, explanations in simpler shapes are easier for the end user to understand; secondly, it is relatively efficient to calculate the volume of a simplex or a box.

The procedure inferConvexCorrection implements the above process using a greedy algorithm. In the case of simplexes, we first randomly choose a discovered region and randomly sample a simplex inside it. Then for each vertex, we move it by a very small distance in a random direction such that (1) the simplex is still contained in the set of discovered regions, and (2) the volume increases. The

process stops until the volume cannot be increased further. For boxes, the procedure is similar except that we move the surfaces rather than the vertices.

Note that our approach is sound but not optimal or complete. In other words, whenever Algorithm 1 finds a symbolic correction, the correction is verified and stable, but it is not guaranteed to be minimal. Also, when our approach fails to find a stable symbolic correction, it does not mean that such corrections do not exist. However, in practice, we find that our approach is able to find stable corrections for most of the time and the distances of the discovered corrections are small enough to be useful (as we shall see in Section 4.2).

### 3.2 Extensions

We finish this section by discussing several extensions to our approach.

**Handling categorical features.** Categorical features are typically represented using one-hot encoding and directly applying Algorithm 2 on the embedding can result in a symbolic correction comprising invalid concrete corrections. To address this issue, we enumerate the embeddings representing different values of categorical features and apply Algorithm 2 to search symbolic corrections under each of them.

**Extending for multiple classes.** Our approach can be easily extended for multiple classes as long as there is only one desirable class. Concretely, we need to: 1) guide the initial concrete correction generation (to the desirable class), which has been studied in the literature of adversarial example generation; 2) extend classConstraints so that the desired class gets a higher weight than any other class. Compared to the binary case, the classConstraints only grows linearly by the number of classes, while the majority of the constraints are the ones encoding the activations. Thus, the time should not grow significantly, as we shall see in subsection 4.2. That said, the focus of our paper is judgment problems which do binary classifications.

**Extending to non-ReLU activations.** Our approach applies without any change as long as the activation functions are continuous and can be approximated using piece-wise linear functions. For networks whose activations are continuous but cannot be approximated using piece-wise linear functions, we can still apply our algorithm but need constraints that are more expressive than linear constraints to represent verified regions. When activations are not continuous, our approach no longer applies as our method of testing whether two regions are connected relies on them being continuous.

**Incorporating prior knowledge on features.** When the user has constraints or preferences over the features, our approach can be extended to incorporate such prior knowledge in the following ways: 1) if some features cannot be changed, we can avoid searching feature combinations involving them, which also saves computationally; 2) if a feature can only be changed to a value in an interval, we simply add this interval as a constraint to LP formulation; 3) if some features are preferable to change, we can adjust the coefficients of features in the distance function accordingly.

## 4 Empirical Evaluation

We evaluate our approach on three neural network models from different domains.

### 4.1 Experiment Setup

**Implementation.** We implemented our approach in a tool called POLARIS, which is written in three thousand lines of Python code. To implement findMinimumConcreteCorrection, we used a customized version of the CleverHans library [24]. To implement isFeasible which checks feasibility of generated linear constraints, we applied the commercial linear programming solver Gurobi 7.5.2 [9].

**Neural networks.** Table 1 summarizes the statistics of the neural networks. The *mortgage underwriting* network predicts whether an applicant would default on the loan. Its architecture is akin to state-of-the-art neural networks for predicting mortgage risks [28], and has a recall of 90% and a precision of 6%. It is trained to have a high recall to be conservative in accepting applications. The *solver performance prediction network* predicts whether a first-order theorem can be solved efficiently by a solver based on static and dynamic characteristics of the instance. We chose its architecture using a grid search. The *drawing tutoring* network judges whether a drawing is an accurate rendition

Table 1: Summary of the neural networks used in our evaluation.

| Application | Network Structure | # ReLUs | Dataset (train/val./test: 50/50/25) | # features | F1 | Accuracy |
|---|---|---|---|---|---|---|
| Mortgage Underwriting | 5 dense layers of 200 ReLUs each | 1,000 | Applications and performance of 34 million Single-Family loans [6] | 21 | 0.118 | 0.8 |
| Solver Performance Prediction | 8 dense layers of 100 ReLUs each | 800 | Statistics of 6,118 first-order theorems and their solving times [13] | 51 | 0.74 | 0.792 |
| Drawing Tutoring | 3 1-D conv. layers (filter shape: [5,4,8]) and 1 dense layer of 1,024 ReLUs | 4,096 | 0.12 million variants of a canonical cat drawing and 0.12 million of cat drawings from Google QuickDraw[8] | 512 | 0.995 | 0.995 |

of a canonical drawing of a cat. A drawing is represented by a set of line segments on a $256 \times 256$ canvas, each of which is represented by the coordinates of its vertices. A drawing comprises up to 128 lines, which leads to 512 features.

**Evaluation inputs.** For the first two applications, we randomly chose 100 inputs in the test sets that were rejected by the networks. For drawing tutoring, we used 100 variants of the canonical drawing and randomly removed subsets of line segments so that they get rejected by the network.

**Algorithm configurations.** Our approach is parameterized by the number of features $n$ allowed to change simultaneously, the maximum number of regions to consider $m$, the stability metric, the distance metric, and the shape of the generated symbolic correction. We set $n = 2$ for mortgage underwriting and solver performance prediction as corrections of higher dimensions on them are hard for end users to understand. Moreover, we limit the mutable features to 5 features each that are plausible for the end user to change. Details of these features are described in Appendix B.1. As for drawing tutoring, we set $n \in [1, 20]$, which allows us to add up to 5 line segments. To reduce the computational cost, we use a generative network to recommend the features to change rather than enumerating all combinations of features. The network is a variational autoencoder that completes drawing sketches [10]. We set $m = 100$ and discuss the effect of using different $m$ later. For the stability metric and the distance metric, we use a weighted $L_\infty$ norm and a weighted $L_1$ respectively for mortgage underwriting and solver performance prediction, which are described in Appendix B.1. For drawing tutoring, we measure the distance of a correction by the number of features changed ($L_0$), which reflects how many lines are added. We say a correction is stable if it contains at least 3 pixels in each dimension. Finally, we use triangles to represent the corrections for mortgage underwriting and solver performance prediction, while we use axis-aligned boxes for drawing tutoring. The blue rectangles in Figure 1(c) are projections of a box correction on coordinates of added line vertices.

**Experiment environment.** All the experiments were run on a Dell XPS 8900 Desktop with 16GB RAM and an Intel I7 4GHZ quad-core processor running Ubuntu 16.04.

### 4.2 Experiment Results

We first discuss how often POLARIS generates stable corrections and how far away these corrections are from the original input. We then study the efficiency. Next, we discuss the effect of varying $m$, the maximum number of regions to consider. We then compare against grid search. Finally, we discuss the performance of POLARIS when there are multiple classes.

**Stability and minimality.** For the selected 100 inputs that are rejected by each network, POLARIS successfully generated symbolic corrections for 85 inputs of mortgage underwriting, 81 inputs of solver performance prediction, and 75 inputs of drawing tutoring. For the remaining inputs, it is either the case that the corrections found by POLARIS were discarded for being unstable, or the case that POLARIS failed to find an initial concrete correction due to the incompleteness of the applied adversarial example generation algorithm. These results show that POLARIS is effective in finding symbolic corrections that are stable and verified.

We next discuss how similar these corrections are to the original input. Figure 2 lists the sorted distances of the aforementioned 85 symbolic corrections. For mortgage application and solver performance prediction, the distance is defined using a weighted $L_1$ norm, where the weight for each feature is 1/(max-min) (see Appendix B.1). The average distances of corrections generated on these two applications are 0.31 and 0.25 respectively. Briefly, the former would mean, for example, to decrease the DTI by 19.5% or to increase the interest rate by 3%, while the latter would mean, for example, to add 25% more unit clauses or horn clauses. Moreover, the smallest distances for these

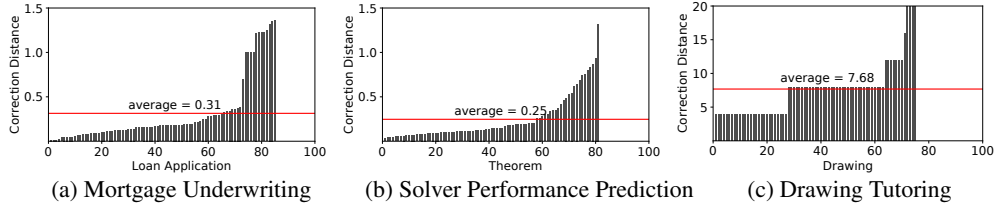

(a) Mortgage Underwriting     (b) Solver Performance Prediction     (c) Drawing Tutoring

Figure 2: Distances of judgment interpretations generated by POLARIS.

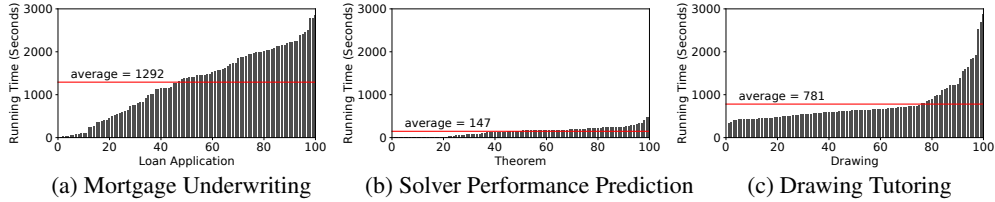

(a) Mortgage Underwriting     (b) Solver Performance Prediction     (c) Drawing Tutoring

Figure 3: Running time of POLARIS on each input.

two applications are only 0.016 and 0.03. As for drawing tutoring, the distance is measured by the number of features to change (that is, number of added lines $\times$ 4). As figure 2(c) shows, the sizes of the corrections range from 1 line to 5 lines, with 2 lines being the average. In conclusion, the corrections found by POLARIS are often small enough to be actionable for end users.

To better understand these corrections qualitatively, we inspect several corrections more closely in Appendix B.2. We also include more example corrections in Appendix B.3.

**Efficiency.** Figure 3 shows the sorted running time of POLARIS across all inputs for our three applications. On average, POLARIS takes around 20 minutes, 2 minutes, and 13 minutes to generate corrections for each input of the three applications respectively. We first observe POLARIS consumes the least time on solver performance prediction. It is not only because solver performance prediction has the smallest network but also because the search often terminates much earlier before reaching the maximum number of regions to consider (m=100). On the other hand, POLARIS often reaches this limit on the other two applications. Although drawing tutoring has a larger network than mortgage underwriting, POLARIS consumes less time on it. This is because POLARIS uses a generative network to decide which features to change for drawing tutoring, which leads to one invocation to Algorithm 2 per input. On the other hand, for mortgage underwriting, POLARIS needs to invoke Algorithm 2 for multiple times per input which searches under a combination of different features. However, a single invocation to Algorithm 2 is still faster for mortgage underwriting.

After closer inspection, we find the running time is dominated by invocations to the LP solver. We have two observations about the invocation time. First, most of the time is spent in instance creation rather than actual solving due to the poor performance of python binding of Gurobi. For instance, in mortgage underwriting, while each instance creation takes around 60ms, the actual solving typically only takes around 1ms. As a result, POLARIS can be made even more efficient if we re-implement it using C++ or if Gurobi improves the python binding. Second, the LP solver scales well as the size of the network and the number of dimensions grow. For example, compared to the solving time (1ms) in mortgage underwriting, where the network comprises 1,000 neurons and the corrections are 2-dimension, the solving time only grows up to around 7ms in drawing tutoring, where the network comprises 4,096 neurons and the corrections are up to 20-dimension. This indicates that POLARIS has the potential to scale to even larger networks with higher input dimensions.

**Varying maximum number of regions.** Table 2 shows the results of varying maximum number of regions to consider ($m$) for four randomly selected inputs of mortgage underwriting. To simplify the discussion, we only study corrections generated under DTI and interest rate. As the table shows, both the volume and running time increase roughly linearly as the number of explored regions grows.

**Comparing to sampling by a grid.** An alternative approach to generate judgment interpretations is to sample by a grid. Since there may be unviable inputs between two adjacent viable inputs, a grid with fine granularity is needed to produce a symbolic correction with high confidence. However, this is not feasible if there are continuous features or the input dimension is high. For instance, the corrections generated on drawing tutoring may involve up to 20 features. Even if we only sample 3 pixels along each feature, it would require over 3 billion samples. Our approach on the other hand, verifies a larger number of concrete corrections at once by verifying a linear region.

Table 2: Effect of varying the maximum number of regions to consider.

| m | # explored regions | volume | time (in seconds) |
|---|---|---|---|
| 100 | 88, 100, 100, 100 | 2.4, 10.3, 9.2, 1.29 | 102, 191, 141, 118 |
| 500 | 88, 205, 214, 500 | 2.4, 26.3, 21.9, 6.9 | 100, 374, 288, 517 |
| 1000 | 88, 205, 214, 1000 | 2.4, 26.3, 21.9, 10.2 | 100, 375, 290, 1115 |
| 2000 | 88, 205, 214, 1325 | 2.4, 26.3, 21.9, 11.2 | 101, 375, 291, 1655 |

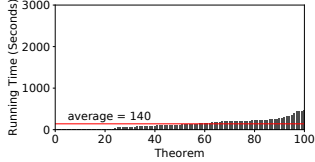

Figure 4: Running time of PO-LARIS for multiple classes.

**Scaling to multiclass classification.** As discussed in Section 3.2, when extending for multiple classes, the runtime of our approach should not grow significantly compared to similar binary class settings. As an empirical justification, we extended the network in our solver performance prediction application to a six-class one. It predicts which solver heuristic out of five can efficiently solve a problem or concludes that none can. To enable a fair comparison, the new network has a similar structure (8 hidden dense layers each of which has 100 ReLUs). We chose one heuristic as the desirable class. Figure 4 shows the sorted running time of POLARIS across the inputs. While it took 147 seconds on average to produce an explanation for the original network, it took 140 seconds for the new network on average.

# 5   Related Work

Our work is related to previous works on interpreting neural networks in terms of the problem [21], and works on generating adversarial examples [7] in terms of the underlying techniques.

Much work on interpretability has gone into analyzing the results produced by a convolutional network that does image classification. The Activation Maximization approach and its follow-ups visualize learnt high-level features by finding inputs that maximize activations of given neurons [5, 12, 17, 31, 23]. Zeiler and Fergus [33] uses deconvolution to visualize what a network has learnt. Not just limited to image domains, more recent works try to build interpretability as part of the network itself [25, 18, 30, 32, 19]. There are also works that try to explain a neural network by learning a more interpretable model [26, 16, 3]. Lundberg et al. [20] and Kindersmans et al. [14] assign importance values to features for a particular prediction. Koh and Liang [15] trace a prediction back to the training data. Anchors [27] identifies features that are sufficient to preserve current classification. Similar to our work, Dhurandhar et al. [4] infers minimum perturbations that would change the current classification. While we infer a stable symbolic corrections representing a set of perturbations, they infer a single concrete correction. As stated in our introduction, it provides many benefits for a correction to be symbolic and stable. In summary, the problem definition of judgement interpretation is new, and none of the existing approaches can directly solve it. Moreover, these approaches typically generate a single input prototype or relevant features, but do not result in corrections or a space of inputs that would lead the prediction to move from an undesirable class to a desirable class.

Adversarial examples were first introduced by Szegedy and et al. [29], where box-constrained L-BFGS is applied to generate them. Various approaches have been proposed later. The fast gradient sign method [7] calculates an adversarial perturbation by taking the sign of the gradient. The Jacobian-based Saliency Map Attack (JSMA) [11] applies a greedy algorithm based a saliency map which models the impact each pixel has on the resulting classification. Deepfool [22] is an untargeted attack optimized for the $L_2$ norm. Bastani at al. [2] applies linear programming to find an adversarial example under the same activations. While these techniques are similar to ours in the sense that they also try to find minimum corrections, the produced corrections are concrete and correspond to individual inputs. On the other hand, our corrections are symbolic and correspond to sets of inputs.

# 6   Conclusion

We proposed a new approach to interpret a neural network by generating minimal, stable, and symbolic corrections that would change its output. Such an interpretation is a useful way to provide feedback to a user when the neural network fails to produce a desirable output. We designed and implemented the first algorithm for generating such corrections, and demonstrated its effectiveness on three neural network models from different real-world domains.

**Acknowledgments**

We thank the reviewers for their insightful comments and useful suggestions. This work was funded in part by ONR PERISCOPE MURI, award N00014-17-1-2699.

## Footnotes

[1]Unless specified, all vectors in the paper are by columns.

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
