[Supplementary Material]

Table 3: Mutable features in corrections to mortgage underwriting.

| | Name | Type | Range | Radius |
|---|---|---|---|---|
| 1 | Interest Rate | Real | [0, 0.1] | 0.005 |
| 2 | Credit Score | Integer | [300, 850] | 25 |
| 3 | Debt-to-Income | Real | [0.01, 0.64] | 0.025 |
| 4 | Loan-to-Value | Real | [0, 2] | 0.025 |
| 5 | Property Type | Category | [Cooperative Share, Manufactured Home, Planned Urban Development, Single-Family Home, Condominium] | N.A. |

Table 4: Mutable features in corrections to solver performance prediction.

| | Name | Type | Range | Radius |
|---|---|---|---|---|
| 1 | % Clauses That Are Unit | Real | [0, 1] | 0.025 |
| 2 | % Clauses That Are Horn | Real | [0, 1] | 0.025 |
| 3 | % Clauses That Are Ground | Real | [0, 1] | 0.025 |
| 4 | Avg. Clause Length | Real | [0, 10] | 0.25 |
| 5 | Avg. Clause Depth | Real | [0, 10] | 0.25 |

# A    Extensions to Our Approach

In this section, we discuss several more extensions to our approach besides the ones discussed in subsection 3.2.

**Extending to other norms.**    If we use norms other than $L_1$ to measure the sizes of vectors, our algorithm largely remains the same, except for $\mathsf{dis}_e$, which measures the stability and size of a inferred symbolic correction. When $L_\infty$ is used, we can still evaluate $\mathsf{dis}_e$ using linear programming. However, when other norms are applied, evaluating $\mathsf{dis}_e$ would require solving one or more non-linear optimization problems.

**Reducing the number of invocations to Algorithm 2.**    When the input dimension is high, Algorithm 1 may lead to a large number of invocations to Algorithm 2 due to the many combinations of different features. One way to avoid this problem is to use another machine learning model to predict which features would yield a desirable correction, as we saw in the cat drawing experiment (Section 4).

**Avoiding adversarial corrections.**    Adversarial inputs are inputs generated from an existing input via small perturbations such that they are indistinguishable to end users from the original input but lead to different classifications. Adversarial inputs are undesirable and often considered as "bugs" of a neural network. For simplicity, we did not consider them in previous discussions. To avoid corrections that would result in adversarial inputs, we rely on the end user to define a threshold $\sigma$ such that any concrete correction $\delta$ where $\|\delta\| > \sigma$ is considered not adversarial. Then we add $\|x\| > \sigma$ as an additional constraint to each region.

# B    Experiment Details

## B.1    Experiment Setup

We describe details about the experiment setup in this subsection.

**Mutable features.**    Table 3 and Table 4 describe the features that are allowed to change in order to generate symbolic corrections for mortgage underwriting and solver performance prediction.

**Stability and distance metrics.**    We first describe the operator $\mathsf{dist}_e$ for the mortgage application, which measures both stability and distance. Briefly, we used a weighted $L_1$ norm to evaluate the

distance of the correction and a weighted $L_\infty$ norm to evaluate the stability. For distance, we use 1 / (max - min) as the weight for each numeric feature. As for the categorical feature "property type", we charge 1 on the distance if the minimum stable concrete correction in the symbolic correction (the minimum stable region center) would change it, or 0 otherwise. This is a relatively large penalty as changing the property type requires the applicant to switch to a different property. For stability, we define a stability radius array $r$ and use $1/r[i]$ as the weight for feature $i$. If the category feature is involved, we require the symbolic corrections to at least contain two categories of the feature. Table 3 defines the range and radius of each feature. We define $\text{dist}_e$ as follow:

$$\text{dis}_e(\Delta) := \min_{\delta \in S}(\frac{|\delta[1]|}{0.1 - 0} + \frac{|\delta[2]|}{850 - 300} + \frac{|\delta[3]|}{0.64 - 0.01}$$
$$+ \frac{|\delta[4]|}{2 - 0} + (0 \text{ if } \delta[5] \text{ leads to no change else } 1)),$$

where

$$S := \{\delta \in \Delta \mid \exists 1 \le i < j \le 4. \forall \delta'. |\delta'[i] - \delta[i]| \le e * r[i]$$
$$\wedge |\delta'[j] - \delta[j]| \le e * r[j]$$
$$\wedge |\delta'[k] = \delta[k]| \text{ for } k \notin \{i, j\}$$
$$\implies \delta' \in \Delta\}$$
$$\cup \{\delta \in \Delta \mid \exists i \in [1, 4] \text{ and a category } c \text{ of Feature 5}$$
$$\text{that differs from the category } \delta[5] \text{ leads to so that}$$
$$\forall \delta'. |\delta'[i] - \delta[i]| \le e * r[i]$$
$$\wedge \delta'[5] = \delta[5] \text{ or } \delta'[5] \text{ leads to } c$$
$$\wedge |\delta'[k] = \delta[k]| \text{ for } k \notin \{i, 5\}$$
$$\implies \delta' \in \Delta\}.$$

Note when the categorical feature property type is involved, we evaluate $\text{dis}_e(\Delta)$ by solving a sequence of integer linear programming problems, which is also implemented using Gurobi.

The definition of $\text{dist}_e$ for solver performance prediction is similar except that all mutable features are real values:

$$\text{dis}_e(\Delta) := \min_{\delta \in S}(\frac{|\delta[1]|}{1 - 0} + \frac{|\delta[2]|}{1 - 0} + \frac{|\delta[3]|}{1 - 0} + \frac{|\delta[4]|}{10 - 0} + \frac{|\delta[5]|}{10 - 0},$$

where

$$S := \{\delta \in \Delta \mid \exists 1 \le i < j \le 5. \forall \delta'. |\delta'[i] - \delta[i]| \le e * r[i]$$
$$\wedge |\delta'[j] - \delta[j]| \le e * r[j]$$
$$\wedge |\delta'[k] = \delta[k]| \text{ for } k \notin \{i, j\}$$
$$\implies \delta' \in \Delta\}.$$

We set $e = 1$ for both applications in all runs in the experiment.

## B.2 Case Study

While the discussion in Section 4.2 gives a high-level idea of the effectiveness of our approach, we now look at individual generated symbolic corrections closely. We are interested in answering two questions:

1. Are these corrections small and stable enough such that they are actionable to the applicant?
2. Do they make sense?

We study corrections generated for mortgage underwriting in detail to answer these two questions. More concretely, we inspect the symbolic corrections generated for the application with the minimum correction among all applications and the ones generated for the application with maximum correction. These two applications correspond to the rightmost and the leftmost bars on Figure 2(a).

Figure 5(a)-(c) shows the symbolic corrections generated for the application with the minimum judgment interpretation among all applications. The application corresponds to the leftmost bar in Figure 2(a). Since POLARIS is configured to generate corrections involving two features out of five features, there are ten possible corrections that vary different features. For space reason, we study three of them.

Figure 5: Corrections for the mortgage application with the minimum judgment interpretation (a,b, and c) and a correction for the mortgage application with the maximum judgment interpretation (d).

Figure 5(a) shows the symbolic correction generated along loan-to-value ratio and property type, which is the minimum correction for this application. The red cross shows the projection of the original application on these two features, while the blue lines represent the set of corrected applications that the symbolic correction would lead to. First, we observe that the correction is very small. The applicant will get their loan approved if they reduce the loan-to-value ratio only by 0.0076. Such a correction is also stable. If the applicant decides to stick to single-family home properties, they will get the loan approved as long as the reduction on the loan-to-value ration is greater than 0.0076. Moreover, they will get similar results if they switch to cooperative share properties or condominiums. This correction also makes much sense, since reducing loan-to-value ratio often means to reduce the loan amount. In practice, smaller loans are easier to approve. Also, from the perspective of the training data, smaller loans are less likely to default.

Figure 5(b) shows the symbolic correction generated along debt-to-income ratio and interest rate, which are two numeric features. Similar to Figure 5(a), the red cross represents the projection of the original application, while the blue triangle represents the symbolic correction. In addition, we use a polytope enclosed in dotted yellow lines to represent the verified linear regions collected by Algorithm 2. We have two observations about the regions. First, the polytope is highly irregular, which reflects the highly nonlinear nature of the neural network. However, POLARIS is still able to generate symbolic corrections efficiently. Secondly, the final correction inferred by our approach covers most area of the regions, which shows the effectiveness of our greedy algorithm applied in inferConvexCorrection. While this correction is also small and stable, its distance is larger than the previous correction along loan-to-value ratio and property type. Such a correction also makes sense from the training data perspective. It is obvious that applicants with smaller debt-to-income ratios will less likely default. As for interest rate, the correction leans towards increasing it. It might be due to the fact that during subprime mortgage crisis (2007-2009), loans were approved with irrationally low interest rate, many of which went into default later.

Figure 5(c) shows the correction generated along debt-to-income ratio and loan-to-value ratio. Compared to the previous corrections, its distance is small but it is highly unstable (the triangle is very narrow). In fact, it is discarded by POLARIS due to this.

As a comparison to corrections generated on the previous application, Figure 5(d) shows the final correction generated on the application that corresponds to the rightmost bar on Figure 2(a). In other words, its final correction has the largest distance among final corrections generated for all applications. As the figure shows, such a large distance makes it hard for the applicant to adopt. For most categories of property type, the applicant needs to raise their credit score by 100, and even to over 800 under some cases, which is not very easy in practice. As a result, POLARIS assigns a high distance for such a correction.

## B.3 More Example Corrections

### B.3.1 Mortgage Underwriting

The red cross shows the projection of the original application on these two features, while the blue lines represent the set of corrected applications that the symbolic correction would lead to. In addition, we use a polytope enclosed in dotted yellow lines to represent the verified linear regions collected by Algorithm 2.

## B.3.2 Solver Performance Prediction

The red cross shows the projection of the original application on these two features, while the blue lines represent the set of corrected applications that the symbolic correction would lead to. In addition, we use a polytope enclosed in dotted yellow lines to represent the verified linear regions collected by Algorithm 2.

### B.3.3 Drawing Tutoring

The red lines boxes are inputs and the blue boxes are the symbolic corrections. Briefly, any sets of lines whose vertices all into them would make the drawing accepted by the neural network. The cyan lines are one example concrete correction in each plot.