[Reviews · NeurIPS 2018]

Reviewer 1



This work proposes a novel method that can potentially provide actionable insight to the user when a neural network makes a less than favorable decision. The paper is interesting in that it provides stable and hence potentially actionable insight that can help the target user change an undesired outcome in the future. The work focuses on asymmetric insight in the sense that insight or suggestions are provided only when the classification is for a certain class. So it is mainly applicable to specific kind of binary classification problems where being classified into one class is more undesirable and requires justification. Some hand wavy arguments are provided in the supplement for extension to multiple classes (one vs all), however it would be good to see experiments on those in practice as it is not at all obvious how the solution extends when you have more than one undesirable class. Technically, they solve a large linear optimization problem to obtain the region of interest based on which the insight is conveyed based on minimal perturbation. Although, they provide some tricks to scale their approach the method seems applicable mainly to smallish or medium sized neural networks (symbolic correction with 8 features is not really that high dimensional). More large scale experiments need to be conducted to justify the solutions they propose. The related work section also is kind of thin. Relevant works on interpretability such as S Lundberg et. al. 2017, Koh et. al. 2017, Kindermans 2018 etc. are not referred to. Also a very relevant work on contrastive explanations which looks for minimal perturbations Dhurandhar et. al. 2018 is not discussed. Focus is on actionability however, its possible that the features they select may not be actionable for every person or instance. For example, interest rate is out of control of the individual or property type is something the individual may be inflexible about depending on his intent (a person wanting to buy a commercial building may not be happy with a recommendation telling to buy a residential condo even if he can afford it as he wants to start a business). So a dicussion with such constraints would be important to have. Minor comment: The text refers to some green lines in figure 1c, but I fail to see any. Post Rebuttal: I thank the authors for their response. I still think this is a borderline paper based on my comments.

Reviewer 2



The authors proposed the concept of "correction for interpretation" and supported it with principles: minimal, stable and symbolic. They suggest that a correction can provide insight on the model. While there are related works investigating the decision boundaries of deep learning models., finding a stable correction seems a novel direction to the best of my knowledge. The authors proposed an algorithm for ReLU based models using existing LP solver. Experimental evaluation on real-world data has been conducted and the results are intuitive to understand. score(out of 5): quality(2) clarity(4), originality(4), and significance(2) Detailed comments: The current method is limited in binary classification. For a common multi-class classification problem, it can be much more (exponential in the number of classes?) challenging and it is unclear that the current method can be easily generalized. Even for a binary classification problem, I would suggest the authors try CTR prediction problem in experiments, which can be more challenging and impactful. The initial region is generated via heuristics, which is one of the main limitations of the proposed methodology. It does not ensure the "minimal" property as the authors noted themselves. The categorical feature is very important and the related content should be placed in the main paper rather than the appendix. The authors missed a line of research on the adversarial targeted attack in deep learning, where people generate slightly perturbed input to fool the model to predict a prespecified label. While the objectives are different, these two tasks could share many similar techniques. There is also a lack of baselines. While this might be a novel application, there can be simple baselines that achieve similar results. For example, a plausible baseline would be selecting the nearest neighbors in the data set that has the target label. Minor comments: Figure 1. The caption should be more descriptive. Line 119: "an an" I have read the rebuttal. While the problem is interesting, the execution for solving the problem can be improved: the authors should provide baselines, even a weak one. Even if the baselines might not obey the criteria strictly but neither does the proposed method. It is difficult to evaluate the contribution of the proposed techniques.

Reviewer 3



This paper addresses explainability for classification problems that use neural networks with ReLU activations. The gist of the approach is to estimate the minimal perturbation to apply to a given instance to flip the output of the classifier. The authors specifically focus on the binary classification problem. They also focus on explaining "negative" decisions or rejections motivated by the need to explain to end users that may be the object of the learning why an AI system is making an unfavorable decision (e.g., denying a loan). This is a very interesting paper but there are two issues that I think should be addressed prior to publications: 1) In the experimental section, a key figure seems to be missing, Figure 2. I have found it very hard to assess how well the technique work. There are little guarantees that the generated interpretations do actually make sense for the end user. I suggest that the authors revisit this question. 2) There is a related piece of work that appeared in February 2018 that I think should be added in the reference list and more importantly discussed or compared with this work: https://arxiv.org/pdf/1802.07623.pdf This reference talks about the generation of contrastive explanations, something that can be interpreted as the missing elements that justify a classification. I recommend doing a thorough comparison with this paper in the related work section. While I understand the argument set forth to only focus on negative predictions, I think that there could be lots of value in explaining also why positive decision are made and how safe they are from becoming negative. That's something that I wished could be explained further or at least addressed in the paper. The algorithm also rely on an input n that represent the number of features that could be tweaked while estimating the distance. Defining this number could be tricky in my view. I also wonder if the authors have considered incorporating prior knowledge on the importance of features to ease some of the computational burden on the algorithm. Finally, while the (popular) ReLU activations are assumed for this work and ingrained in the algorithm (exploiting the fact that these activation functions are piece-wise linear), is it possible to extend this work beyond these activation functions?